# Has the COVID-19 Pandemic Cooled Down or Stimulated the Countertendencies of Capital? A Critical Review

Leonardo Carnut [1], Lucas Uback [2,*] and Áquilas Mendes [3,4]

1   Centro de Desenvolvimento do Ensino Superior em Saúde, Universidade Federal de São Paulo, São Paulo 04039-032, Brazil
2   Faculdade de Medicina, Universidade Municipal de São Caetano do Sul, São Caetano 09521-160, Brazil
3   Faculdade de Saúde Pública, Universidade de São Paulo, São Paulo 01246-904, Brazil
4   Programa Pós-Graduados em Economia, Pontifícia Universidade Católica de São Paulo, São Paulo 05014-901, Brazil
*   Correspondence: uback03@gmail.com

**Abstract:** This is a critical review of what the Marxist scientific literature presents on the forms of countertendency to falling profit rates carried out during the COVID-19 pandemic. The 33 articles included in this review were studied using a Marxist approach. The following elements of the articles were synthesized and criticized: the analysis matrices, the methodological aspects of the articles, the elements contrary to the law of the tendency of the rate of profit to fall, and the cases studied and their contexts of analysis. The articles reviewed allow us to state that during the COVID-19 pandemic, there was intensification in the forms of an increased degree of labour exploitation, cheapening of the elements of constant capital, and increased relative overpopulation and shareholder capital.

**Keywords:** coronavirus; criticism; capitalism; review

## 1. Introduction

The current historical moment is marked by one of the biggest and longest crises of the capitalist system, crossed by the pandemic of the new coronavirus. This demands from the working class the need to deeply understand the multiple determinations that rule capitalism as it is configured today. The appropriation of this knowledge is a condition for the radical social transformation to be operated by this class in the course of history. For that, the historical–dialectical materialism, a method developed by Karl Marx and Friedrich Engels exposed in the text *The 1857 introduction–The method of political economy*, offers the path to investigate the concrete reality and overcome this type of social organization based on the exploitation of the human being by the human being.

It is known that the existence of capitalism is marked by crises, generated by specific conditions created by the dynamics of the capitalist system itself. In turn, it cannot be ignored as a characteristic of capitalism, according to Munger and Villarreal-Diaz (2019), that clientelism is intrinsic and inseparable from capitalism, which can reinforce its process of crisis instability. According to Manzano (2013), the crisis arises from disorderly production and the contradiction generated between the extension of consumption and profit realization. This is because in order to increase mass consumption, it would be necessary to increase wages, reducing the rate of surplus value appropriated by the bourgeoisie, and consequently, its profits. To overcome this situation, market expansion is a constant pursuit, intensifying the terms of the contradiction exposed.

Fontes (2017, p. 414) alerts that "crises are the genetic mark of capitalism" and highlights some of its nuances: overproduction, generated by the goods that were not converted into merchandise; profitable destruction, as in the case of wars and the consequent expansion of the war industry; imposition of the replacement of necessary goods already acquired, given the reduction in the time of use; new types of expropriations and rapine,

such as public debt; rupture of the social metabolism between the social being and nature; and the emptying of human relations, violence, mental illness, and other harmful effects produced by crises.

In contemporary times, capital, as Fontes highlights, has its main role in the financialization process. According to the author, "financialization should not be considered as a power of money, isolated from the processes of extracting value" (Fontes 2017, p. 416) but tied together to the same contradictory correlation between 'pure property', extraction of value, and expropriations on which capitalists base the processes of value extraction on the working class. This is partially supported by the role of the State, especially in the existence of an ongoing trend in recent decades characterized by the withdrawal of rights and setbacks in the achievements of workers, baptized by Fontes (2010) as 'secondary expropriation'.

As Mendes and Carnut (2018) point out, the current capitalist crisis began in the 1970–80s and is still dragging on, and should be understood from two main trends articulated among themselves: (a) the falling tendency of the rate of profit with certain anemic recoveries that characterize the path of capitalist decadence, and (b) the dominance of interest-bearing capital in its form of fictitious capital (financial capital). Regarding the second tendency, Meramveliotakis (2022), when referring to this domain of fictitious capital, specifies the treatment of this category in Marx, saying that it contributes to the reflection on the financialization process particularly present in the 21st century. For this author, finance plays a contradictory role in capitalism. It is perceived that finance and the financial markets are fundamental instruments in supporting the process of capital accumulation. Undoubtedly, it constitutes an important accelerator of the accumulation process, accelerating the pace of accumulation, which is the hallmark of the current dynamics of capitalism.

The existence of tendencies, in turn, is accompanied by their opposite, by countertendencies, whose action takes place in the sense of "overcoming or delaying the effect of generative mechanisms originating in objects" (Paulo 2020, p. 7). These result from the gap between the praxis and the conscious action of individuals in the process of their objectification (Paulo 2020). Countertendencies do not erase the existence of laws, but prevent them from having an absolute character; in this sense, their recognition and study enable the working class to advance in the conscious production of alternatives to counter the capital's tendency movements, and ultimately, build a society that has as its horizon the dissolution of social classes.

In *The Capital*, *Volume III*, Marx announces the tendency of the rate of profit to fall. The rate of profit is expressed as the result of the ratio between surplus value and total capital invested, which in turn is composed of a constant part (means of production) and a variable part (employed labour power). The capital expansion movement, by keeping constant the degree of exploitation of labour, promotes the increase in the constant share of total capital in proportion to its variable share, necessarily resulting in a gradual fall in the general rate of profit (Marx 2017b). There are six countertendencies that accompany the tendency of the rate of profit to fall described by (Marx 2017b): (1) increase in the degree of exploitation of the labour force; (2) compression of salaries under their value; (3) cheapening of the elements of constant capital; (4) relative overpopulation; (5) foreign trade; and (6) increased shareholder capital.

Still, it is necessary to emphasize that capital faces, nowadays, in addition to the structural crisis, a serious health crisis caused by the new coronavirus. The COVID-19 pandemic claimed more than 6.32 million human lives so far (Johns Hopkins University and Medicine 2022) and is not unrelated to the pattern of (re)production of life under the aegis of capitalist social relations. On the contrary, large-scale agribusiness acts in the creation and propagation of new diseases, either through the creation of new pathogens or through the immunopathological disruption of ecosystems (Wallace 2020; Carnut et al. 2021). Global-scale contagion is associated with the large circulation of people, food, and goods in an increasingly shorter time and space (Wallace 2020). Even more, Fasfalis (2020) argues that the COVID-19 pandemic reveals that the productive forces accumulated on a world scale turned into forces of destruction that lead very deeply to postmodern barbarism.

To face this, even in terms of health care, Lyon-Callo (2020) asks Richard Wolff, and receives this answer:

> It is not profitable for companies to produce a mask or a bed or a glove. To produce these things, to store them in some warehouse, let alone to stockpile them all over the country, waiting who knows how many months for the next virus to show up, is not profitable. The risk is enormous. You are just not going to do it as a capitalist. You can find more profitable, less risky investments elsewhere. How do we know that? Because that is what they did. They did not make the stuff, and we were not prepared. (Lyon-Callo 2020, p. 571)

As Pereira and Pereira-Pereira (2021) emphasize, the COVID-19 pandemic should not be understood as an isolated fact. It is an event that was predicted in recent years, even foreseen by the World Health Organization (Carnut et al. 2020) and, for this reason alone, should not come as some surprise. It is in this context that we can affirm that the COVID-19 pandemic is located in history as one of the products of capital in crisis, at the same time that it imposes on it a new dynamic for the realization of its own relations of exploitation and expropriation. Nelson's (2020) argument is interesting, in which he mentions that COVID-19 presents itself as a different order of crisis. According to this author, in reality, this calamity really exposes the fatal weaknesses of capitalism. Bhattacharya and Dale (2020) aggregate that COVID-19 starkly revealed not only the brutal systemic priorities of capitalism—profit-making over life-making—but also the relationship between capital and the capitalist state form.

Although the thesis is that the COVID-19 crisis inevitably accentuated or amplified these contradictions within capitalism, some authors suggest that it is "obvious" (Baski 2020). However, it is still not well mapped. Even when we agree with the fact that COVID-19 represents a conjunctural crisis, it still remains to be seen how the countertrends to this crisis are (or are not) in line with the fundamental contradictions of the capitalist dynamic.

Given the above information, it was identified that there is a need to better understand the attempts that capitalists used to counteract their losses in this scenario in which COVID-19 is a byproduct of their own greed and overexploitation. For this purpose, this study aims to review the forms of countertendency to the falling profit rate effected during the COVID-19 pandemic identified in recent Marxist scientific literature.

## 2. Outlining the Review Path

It was a critical review (Grant and Booth 2009; Gough et al. 2012), focusing on the Marxist scientific literature in line with that carried out by Carnut (2022), which was guided by a research question, defined as 'what does the Marxist scientific literature present about the forms of countertendencies to the law of the tendency of the rate of profit to fall during the COVID-19 pandemic?'. The question allowed for the delimitation of some free terms used to perform the search in the listed data sources.

The bibliographic search of the studies was initially carried out in 55 Brazilian, Latin American, American, European, and Australian Marxist journals. It was considered relevant to include annals of scientific events that disseminate Marxist content, promoted by the Interdisciplinary Centre for Studies and Research on Marx and Marxism (NIEP-Marx), the Brazilian Society for Political Economy (SEP), and the Latin American Political Studies Group (GEPAL). It is important to highlight that, even considering outspokenly Marxist journals (as exposed in their scope), it is possible to identify a certain degree of heterodoxy in some of them, allowing the publication of other perspectives (such as the Keynesian one for example) in the dialogue with the Marxist.

The free terms derived from the research question were grouped into two poles. The first (pole 1) related to the 'law of the tendency of the rate of profit to fall' (LTRPF), whose terms were 'countertendency'; 'tendential law'; 'falling tendency of the rate of profit'; 'fall in the rate of profit'; and the 'law of the falling rate of profit'. The second (pole 2) related to the

COVID-19 pandemic had the following free terms: 'COVID-19'; 'coronavirus pandemic'; 'new coronavirus'; and 'SARS-CoV-2'.

Three stages of identification of publications were carried out in the selected data sources. In the first stage, the free terms without quotes and their respective translations were searched in each of the data sources (when the journal published only in English or Spanish). The second stage was carried out with the sources that exceeded 50 retrieved publications, so the search was conducted again using quotation marks in an attempt to refine the results. In the first two stages, the source eligibility criterion was the presence of at least one free term from each of the poles. A third step was carried out with the sources eligible in the first step, and the search was conducted again by combining the terms of pole 1 and pole 2 using the Boolean operator "*AND*".

At the end, it was possible to identify 1049 publications distributed in 30 journals and 1 annal of scientific event produced between 2020 and 2022. Subsequently, 695 repeated titles were excluded. In the screening phase, publications of bibliographic materials other than in the scientific article format (reviews, editorials, protocols, book chapters, entire volume, and collaborators' notes—23 publications) and those that were unrelated to the theme (288 publications) were excluded. The relation with the theme was identified from the presence of the following textual markers: 'capital', 'profit', 'rate', 'fall', 'law', 'tendency', 'neoliberal', 'capitalism', 'crisis', 'pandemic', 'coronavirus', and 'COVID-19'. At the eligibility stage, among the selected articles, 10 articles were unavailable for full reading. At the end, 33 articles were considered as included in this review. This search was conducted from February 1st to 4th March 2022.

Finally, the historical–dialectical materialist method was used as an exposition method, as approached by Müller (1982), in the following demonstrative construction sequence: exposition, progressive–regressive procedure, contradiction, and criticism. The fundamental concept in Marx is that of "exposure", which designates the way in which the object, sufficiently apprehended and analyzed, unfolds into its own articulations and how thought develops them in their corresponding conceptual determinations, organizing a methodical discourse (Müller 1982; Collin 2006).

The 33 articles included in the review were read in their entirety, and the following information was extracted from their content: author(s); analysis matrices; country of origin; methodology; elements contrary to LTRPF, according to Marx's (2017b) description in chapter 14 of *The Capital Volume III*; and case and context. Table 1 presents the synthesis of such content, allowing a comparative analysis of the results and an overview of their relationship with the theme of this review.

**Table 1.** Articles included by author, year, analysis matrix, country, case, context, and elements contrary to the LTRPF in the period of the COVID-19 pandemic, February–March 2022.

| Author, Year | Analysis Matrix | Country | Case | Context | Elements Contrary to the Law of the Tendency of the Rate of Profit to Fall Identified |
|---|---|---|---|---|---|
| Melim and Moraes (2021) | Marxist | Brazil | ■ Teachers from the Laureat Education Network were replaced by robots<br>■ More than 300,000 dismissals of teachers from the private education network, who served children aged 0–5 years;<br>■ There were more than 1800 dismissals of higher education professors until August/2020 in São Paulo;<br>■ Professors at the University of Guarulhos denounced a contractual amendment that provided for professionals to cede all content produced remotely during the pandemic to the institution, indefinitely and without remuneration. | Distance learning | ■ Cheapening of the constant capital elements;<br>■ Increased degree of labour exploitation. |

**Table 1.** *Cont.*

| Author, Year | Analysis Matrix | Country | Case | Context | Elements Contrary to the Law of the Tendency of the Rate of Profit to Fall Identified |
|---|---|---|---|---|---|
| Bispo and Caldeira (2021) | Marxist | Brazil | ▪ Unemployment was higher among the most precarious workers (blacks, indigenous people, people living in the suburbs, informal workers, women, especially black women), and potentially reached 40 million; ▪ State release of BRL 1.216 trillion for the financial market, and BRL 123.9 billion for emergency aid; ▪ Precarization of work in vulnerable groups, especially black women. | 2007–2008 Crisis | ▪ Increased degree of labour exploitation; ▪ Increased shareholder capital. |
| Araujo et al. (2021) | Keynesian | Brazil | ▪ Fiscal austerity policies at first; ▪ Tax subsidies in the productive and banking sectors; ▪ Flexibilization of labour relations. | 2007–2008 Crisis | ▪ Increased shareholder capital; ▪ Increased degree of labour exploitation. |
| Pereira and Puchale (2021) | Marxist | Brazil | ▪ Unemployment; ▪ Decreased income from work; ▪ Increased informality rate and underemployment. | Setbacks in the labour market (2014) and coup (2016) | ▪ Increased degree of labour exploitation; ▪ Increased relative overpopulation; ▪ Cheapening of the constant capital elements. |
| Silva and Silva (2021) | Marxist | Brazil | ▪ Social isolation increased free reproduction and care activities, performed mostly by women, mainly black; ▪ Fiscal austerity policies, public spending cuts, and the dismantling of strategic Ministries that protect the working class. | Neoliberalism and dismantling of social protection mechanisms | ▪ Increased degree of labour exploitation. |
| Sabino and Alves (2021) | Keynesian | Brazil | ▪ Provisional measures and executive decrees that intensify work precarization. | "Labour reform" (Law no. 13.467/2017) | ▪ Increased degree of labour force exploitation. |
| Mattei and Heinen (2021) | Marxist | Brazil | ▪ Decline in the unemployment rate reached the highest level in the historical series; ▪ Massive exit of 10 million Brazilians from the labour market and reduction in the number of hours worked; ▪ Unemployment potentially reached 35.6% by the second half of 2020; ▪ Contraction in the income of employed workers. | Slowdown in 2019 | ▪ Increased degree of labour exploitation; ▪ Increased relative overpopulation. |
| Alencar (2021) | Marxist | Brazil | ▪ The amount of federal public expenditures grew between 2015 and 2020, with emphasis on financial expenses; ▪ Social spending lost relative importance in relation to other non-financial spending, becoming important with COVID-19. | Counter-reforms, new fiscal regime (EC-95) | ▪ Increased shareholder capital. |
| Lima et al. (2021) | (Neo)-marxist | Brazil | ▪ Reduction in the labour market participation of women, especially black women; ▪ Worsening inequalities between men and women, and between white and non-white women; ▪ Greater impact for less-skilled and lower-income workers. | COVID-19 shutdown of non-essential activities | ▪ Increased degree of labour force exploitation; ▪ Increased relative overpopulation. |
| Sant'ana and Montoya (2021) | Marxist | Brazil | ▪ Latin American immigrant workers receive low wages, despite their professional qualifications, experiencing a process of "uberization", without minimum labour protection; ▪ Food delivery companies by app grew their profits during the pandemic; ▪ Expansion of precarious labour relations among Latin American immigrants. | The precarization and overexploitation of immigrant labour | ▪ Cheapening of the constant capital elements; ▪ Increased degree of labour exploitation. |
| Ribeiro (2021) | Marxist | Brazil | ▪ Tendency to deepen the economic crisis that dragged on since 2008–2009; ▪ Exposure of ecological and epidemiological disruptions caused by capitalism; ▪ Intensification of the international hegemony disputes between the United States and China. | Neo-liberalism and the preponderance of finance capital | ▪ Increased degree of labour exploitation; ▪ Increased shareholder capital. |

**Table 1.** *Cont.*

| Author, Year | Analysis Matrix | Country | | Case | Context | | Elements Contrary to the Law of the Tendency of the Rate of Profit to Fall Identified |
|---|---|---|---|---|---|---|---|
| Lima (2021) | Keynesian | Brazil | ■ | Extraordinary policies to support the financial system, companies, employment, and income used by the State. | 2020 Crisis | ■ ■ | Increased degree of labour exploitation; Increased shareholder capital. |
| Saad-Filho (2020) | Marxist | United Kingdom | ■ ■ | Central banks provided direct financing for large companies; In several countries, testing for COVID-19 was restricted to health workers, who were dealing with an overload of work and without adequate protection. | State, neo-liberalism, deindustrial-ization and "globalization" | ■ | Increased shareholder capital. |
| Fazzari and Needler (2021) | Post-keynesian | USA | ■ ■ ■ ■ ■ ■ ■ | Unlike the Great Recession (2007–2009), in the COVID-19 pandemic, women were the most affected by unemployment; White workers perform better than Asians, blacks, and Hispanics in both recessions; Black and Hispanic women were hit hardest by the COVID-19 pandemic; Young workers suffer job losses from jobs in both recessions; Middle-aged workers were less severely affected by COVID-19; Older workers fared worse with COVID-19; Workers with low education suffer more from unemployment. | Great Recession and women's work | ■ ■ | Increased degree of labour force exploitation; Increased relative overpopulation. |
| Bortz et al. (2020) | Keynesian | Argentina | ■ ■ ■ | Transfer of public resources to the productive and financial sector via fiscal and budget packages; 25% of measures to support social assistance programs were implemented in a timely manner; Fiscal pacts in EMDEs were less than half of packages in advanced economies. | Capital flight from emerging economies to major financial centers | ■ | Increased shareholder capital. |
| Vernengo and Nabar-Bhaduri (2020) | Keynesian | USA | ■ ■ ■ ■ | Countercyclical fiscal policies in action were strengthened; Purchase of long-term treasury bonds by the Fed; The Central Bank, to a certain extent, resumed its role as a fiscal agent of the state, an important role during the 1930s; Women were most affected with unemployment and child care burden. | Great Recession (2007–2009) | ■ ■ | Increased degree of labour force exploitation; Increased shareholder capital. |
| Baines and Hager (2021) | Post-Keynesian Marxist | USA | ■ | The Fed and the Treasury supported the debt market with investments. The largest corporations continued to use leverage as a form of power, borrowing at low cost and enriching shareholders through dividends and buybacks. | Debit service | ■ | Increased shareholder capital. |
| Canelli et al. (2021) | Post-keynesian | Italia | ■ ■ | European Union domestic policies can slowly halt the decline in GDP and employment and temporarily delay a further increase in public debt in relation to GDP; Government finances, GDP, and employment rate will not return to pre-COVID-19 levels, nor will they reach more sustainable long-term levels. | 2007–2008 Global financial crisis | ■ | Increased shareholder capital. |
| Storm (2021) | Keynesian | The Netherlands | ■ ■ ■ | The United States and the United Kingdom, despite concentrating most of the deaths from COVID-19, did not increase their spending on fighting the pandemic beyond the average; Germany, Canada, Japan, and New Zealand, which could did increase spending on COVID-19 relief by more than average, had below-average mortality rates; More fiscally constrained countries, such as France, Italy, Portugal, and Spain, which were unable to increase additional spending to the same extent, had above-average mortality rates. | Fiscal austerity and neo-liberalism | ■ | Increased shareholder capital. |

**Table 1.** *Cont.*

| Author, Year | Analysis Matrix | Country | Case | Context | Elements Contrary to the Law of the Tendency of the Rate of Profit to Fall Identified |
|---|---|---|---|---|---|
| Banfield (2021) | Neo-classical | USA | ■ Student fees and costs for teaching materials continue to rise;<br>■ Problems with using open online educational resources: costs for creation and adaptation, not all elements of a course can rely on online educational resources, and the quality of such resources already available is not always good;<br>■ A cost-sharing and profit-sharing model would provide long-term sustainability to the development of high-quality, low-cost curricula. | Higher education | ■ Cheapening of the constant capital elements. |
| Stewart et al. (2020) | Marxist | Brazil France United Kingdom | ■ In April 2020, there was a 171% increase in the area of destroyed forest, compared to April 2019;<br>■ Bolsonaro challenged the Federal Court's order to restrict illegal logging and mining in slash and burn areas;<br>■ 25% reduction in IBAMA's budget and dismissal of its director;<br>■ Video of Bolsonaro and his ministers meeting, in which Ricardo Salles suggests that the government take advantage of the press attention focused on the pandemic to relax regulations in the Amazon. | Deepening of subservience to agribusiness | ■ Cheapening of the constant capital elements. |
| Šumonja (2020) | Marxist | Serbia | ■ Keynesian-inspired policies implemented by states during COVID-19 lead to a misconception that we are witnessing "the death of neo-liberalism";<br>■ The 'reversion of the State' is an ideological misconception that hides neoliberal restructuring to restore the rate of profit. | State as an organizing force of the neo-liberal attack | ■ Cheapening of the constant capital elements;<br>■ Increased shareholder capital. |
| Lust (2021) | Marxist | Peru | ■ Those most affected by COVID-19 are working-class people;<br>■ Decreased government revenue and increased spending on health and financial assistance for the most vulnerable families;<br>■ Closing of micro-enterprises, medium-sized enterprises, and large corporations;<br>■ Micro-enterprise workers were laid off directly. Workers from medium and large corporations maintained their wages or were laid off, temporarily laid off, or had their working hours reduced;<br>■ Most private sector workers have temporary contracts, which ended without any possibility of legal challenge;<br>■ Growth of informal work;<br>■ Granting of loans to 17.3% of all micro- and small enterprises. | Peru implemented strict quarantine measures, neoliberal extractive development model | ■ Increased degree of labour force exploitation. |
| Kiliç (2020) | (Neo)-marxist | Turkey | ■ Unemployment and inequality increased during the pandemic, laying a solid foundation for an ideological transformation and a turning point at the end of the neoliberal "long wave";<br>■ COVID-19 generated sympathy for Keynesian measures;<br>■ Trade unions have more arguments to persuade the capitalist state for a more regulated labour market this time around. However, it is necessary to encourage unions to take collective political action;<br>■ Liberal and conservative capitalists are likely to claim that more flexibility will solve the unemployment problem after COVID-19. | Neo-liberalism and the global financial crisis | ■ Not specified |
| Paulsson and Koglin (2022) | Keynesian | Sweden | ■ The number of passengers using public transport in Stockholm dropped dramatically and about 1 billion Swedish kronor per month was lost;<br>■ Incentive contracts were temporarily replaced by a production-based remuneration model;<br>■ While the norms underpinning commoditization remained unquestioned, the techniques were partially reevaluated in the midst of the global crisis as a way to protect the established neoliberal policies. | Introduction of market mechanisms in Stockholm's public transportation | ■ Increased degree of labour exploitation;<br>■ Cheapening of the constant capital elements. |

Table 1. *Cont.*

| Author, Year | Analysis Matrix | Country | Case | Context | Elements Contrary to the Law of the Tendency of the Rate of Profit to Fall Identified |
|---|---|---|---|---|---|
| Dean et al. (2021) | Keynesian | Australia | ■ Income tax cuts, business subsidies and a more employer-friendly labour relations 'reform'; <br> ■ Australia should: (a) ensure full domestic production of some products (defense, energy, and health products); (b) achieve self-sufficiency in the manufacturing sector, expanding it by almost 50%, and (c) increase domestic production of manufactured goods connected to renewable energy. | Dependence on the Austalian economy | ■ Cheapening of the constant capital elements. |
| Żuk and Żuk (2022) | (Neo)-marxist | Poland | ■ COVID-19 marks a fourth wave, with job and salary cuts, precarious employment contracts, and reduced overtime rates; <br> ■ Signing of employment contracts brokered by agencies facilitated such changes; <br> ■ Precarious employment contracts often reduce occupational hygiene and safety; <br> ■ Increased indebtedness and multiemployment of the precariat. | Waves of labour precarization in Poland | ■ Increased degree of labour exploitation. |
| O'Keeffe and Papadopoulos (2021) | (Neo)-marxist | Australia | ■ Australia implemented the 'JobKeeper' and 'JobSeeker' stimulus packages; <br> ■ The 'JobKeeper' excluded workers who were in the sectors most impacted by shutdowns during the pandemic and who have high precarization rates; <br> ■ The 'JobMaker' suspends guarantees of no salary reduction and maintenance of working conditions; <br> ■ Business demands are positioned in the discourse as beneficial to society and workers. | Flexibilization of labour laws in Australia | ■ Increased degree of labour exploitation; <br> ■ Increased relative overpopulation. |
| Van Barneveld et al. (2020) | Keynesian | Australia | ■ Isolation measures have a more negative impact on the most vulnerable population; <br> ■ Increase in unemployment, especially among women, migrants, and workers with precarious contracts; <br> ■ Decline in tourism and traveling; <br> ■ Improved air quality in some of the most polluted places in the world; <br> ■ USA and Australia suspended environmental protection laws; <br> ■ Greater economic devastation among Southern European countries compared to the North. | Multilaterial bodies and neoliberalism | ■ Cheapening of the constant capital elements; <br> ■ Increased degree of labour exploitation; <br> ■ Increased relative overpopulation. |
| Colley et al. (2021) | Post-keynesian | Australia | ■ In Australia, the response to the pandemic so far is more moderate than the response to the GFC, with fewer partisan differences between jurisdictions; <br> ■ In Australia, senior executive service employees were more affected than lower-level employees during the GFC, but faced a similar impact during COVID-19; <br> ■ Delay of previously negotiated wage increases; <br> ■ Postponement of the negotiation of new agreements. | Global financial crisis (GFC) | ■ Increased degree of labour exploitation. |
| Jenkins and Smith (2021) | Keynesian | Australia | ■ Economic and political common sense paved the way for seeing the home and its productivity as fully appropriable. | Lockdown | ■ Cheapening of the constant capital elements; <br> ■ Increased degree of labour exploitation. |
| Benanav (2021) | Marxist | Germany | ■ Pandemic lockdowns hit the service sector workforce hardest, especially low-productivity, low-wage jobs; <br> ■ Pandemic unemployment is particularly bad for women; <br> ■ Spending by peripheral countries to fight the pandemic is lower due to the restrictions imposed by previously accumulated public debts. | World urban population and reconfiguration of labour force exploitation | ■ Increased degree of labour exploitation. |
| Carnut et al. (2020) | Marxist | Brazil | ■ President Bolsonaro shows disregard for the extent of the harm of COVID-19, encourages disrespect for social isolation, and presents the real intention of exposing the working class to the risk of contagion, decimating the most vulnerable part of this class; <br> ■ Low resources allocated to fight the coronavirus until 12 May 2020: USD 1.5 billion (all in US dollars), or 5.4% of the total budget of the Ministry of Health for 2020; <br> ■ The government develops a fascist rhetoric to deal with the problems of the current crisis, placing the ability to solve the problems on individuals. | Bolsonaro in Brazil | ■ Increased degree of labour exploitation. |

Source: elaborated by the authors.

The included articles presented diverse results that gradually approach the research question, whether on the subject of countertendencies to LTRPF or in relation to the effects

of the COVID-19 pandemic. Most of them sought to understand the effects of the new coronavirus pandemic on society, describing and analyzing its impacts on the working class, public policies implemented by the State in various sectors, and actions taken by the private sector. The different methodological approaches and the different philosophical paradigms used by the authors allowed for a multifaceted study of the theme in question.

### 3. Central Aspects of Studies on the Relationship between Countertendencies and COVID-19

#### 3.1. Cases Studied

It is noted that labour relations, unemployment, and government actions to combat the COVID-19 pandemic were themes present in most of the articles in this review. The group of articles dealing with labour relations analyze the implementation of remote work (Benanav 2021; Jenkins and Smith 2021; Melim and Moraes 2021), labour force employment regimes (Żuk and Żuk 2022), public–private partnership regimes (Paulsson and Koglin 2022), reduction in working hours and income (Lust 2021), "uberization" of work (Stewart et al. 2020), precariousness of female work (Van Barneveld et al. 2020; Bispo and Caldeira 2021; Lima et al. 2021; Silva and Silva 2021), worker indebtedness (Żuk and Żuk 2022), and wage arrears (Colley et al. 2021). The issue of unemployment appears in cases of replacement of the labour force by the incorporation of technology in the jobs that characterize the population profile of unemployed workers (Bispo and Caldeira 2021; Fazzari and Needler 2021; Lima et al. 2021; Melim and Moraes 2021; Pereira and Puchale 2021; Vernengo and Nabar-Bhaduri 2020), measures of the occupancy rate (Mattei and Heinen 2021), and increased informal work (Lust 2021).

Regarding the actions of the State in the face of the COVID-19 pandemic, the cases analyzed were about: austerity policies and precarious living conditions of the working class (Carnut et al. 2020; Sabino and Alves 2021; Silva and Silva 2021; Storm 2021), comparison of mortality rates between countries depending on the type of economic policy adopted (austerity and social spending expansion) (Storm 2021), ideological discourses to support the requalification of the exploitation of the labour force (O'Keeffe and Papadopoulos 2021), economic packages (Araujo et al. 2021; Canelli et al. 2021; Lust 2021; O'Keeffe and Papadopoulos 2021), public–private partnerships (Paulsson and Koglin 2022), Keynesian-inspired economic measures (Šumonja 2020; Vernengo and Nabar-Bhaduri 2020), deregulation of environmental protection laws (Ribeiro 2021; Stewart et al. 2020), debt purchases and other forms of transfer of public resources to the private sector (Baines and Hager 2021; Bortz et al. 2020; Bispo and Caldeira 2021; Lima 2021; Ribeiro 2021; Saad-Filho 2020), and the amount of State social expenditure (Alencar 2021; Bispo and Caldeira 2021).

This classification does not include works by Dean et al. (2021), whose case under analysis is a new model of economic policies for Australia, which do not fit into this classification; by Banfield (2021), who proposes a cost-sharing and profit-sharing model for teaching materials used in higher education; and by Kiliç (2020), who analyzes the current situation as a possibility for the end of the neoliberal "long wave".

#### 3.2. Case Analysis Context

The context of the cases exposed in each article contains heterogeneous elements, which together provide an overview of the current capitalist crisis and its relationship with COVID-19. It is noted that most authors draw comparative parallels between the current situation and the 2007–2008 crisis as if, in historical terms, the 2007–2008 crisis and the coronavirus crisis were contiguous (Kiliç 2020; Araujo et al. 2021; Bispo and Caldeira 2021; Canelli et al. 2021; Colley et al. 2021; Fazzari and Needler 2021; Vernengo and Nabar-Bhaduri 2020). Another significant portion discusses the implications of neoliberalism and austerity policies for the dimension of the current crisis of the COVID-19 pandemic (Van Barneveld et al. 2020; Kiliç 2020; Lust 2021; Ribeiro 2021; Saad-Filho 2020; Silva and Silva 2021; Storm 2021). Other contextual elements are: the impacts of social isolation on the organization of work (Jenkins and Smith 2021; Lima et al. 2021); the forms of labour

precarization and overexploitation (Benanav 2021; Sant'ana and Montoya 2021); changes in women's work (Fazzari and Needler 2021; Silva and Silva 2021); distance learning (Melim and Moraes 2021); capital flight from emerging economies to large financial centers (Bortz et al. 2020); state indebtedness and consequent debt service (Baines and Hager 2021); and the deepening of subservience to agribusiness (Stewart et al. 2020).

The specificities of some countries should be noted. In Poland, Żuk and Żuk (2022) identify three waves of labour precarization, the last one being triggered under the occurrence of the COVID-19 pandemic. In Australia, the pre-pandemic context was marked by the commodification of public services, such as the transport sector (Paulsson and Koglin 2022), flexibilization of labour laws (O'Keeffe and Papadopoulos 2021), and economic dependency (Dean et al. 2021). In the Peruvian case, Lust (2021) highlights that even in the face of an extensive blockade of a movement of people and strict quarantine measures, the working class was quickly contaminated and became ill. He explains that this phenomenon derives from Peru's peripheral position in the international division of labour, as well as adoption of a neoliberal extractive development model and of public policies based on neoliberal principles.

The similarity of Brazil to its Latin American neighbor is not a coincidence, but rather engendered by its subordinate position to the capitalist core countries. Neoliberal policies were adopted more intensively in mid-2014 and signed with the parliamentary coup of 2016 that introduced, in addition to other measures aligned with big business, an extremely austere new fiscal regime and a labour reform that set back the achievements of workers (Alencar 2021; Pereira and Puchale 2021; Sabino and Alves 2021). The neo-fascist context that allowed the execution of even more aggressive measures against the working class is explained by Carnut et al. (2020).

### 3.3. Analysis Matrices

Regarding the theories used to support the analyses carried out by the studies, there was a plurality that can be systematized into six groups.

The first group is composed of 10 articles that adopted the '*Keynesian*' paradigm as an analytical reference. This paradigm, established within the liberal tradition, began to take shape in the decades following the First World War (1914–1918), especially with the publication of *The General Theory of Employment, Interest and Money* by John Maynard Keynes in 1936 (Montaño and Duriguetto 2011). It emerges, therefore, in a context marked by successive capitalist crises (with emphasis on the crisis of 1929), sudden drops in the rate of profit, and consolidation of labour organizations (stating the constitution of the working class as a class for itself), to name a few of the cases in progress during this period (Montaño and Duriguetto 2011). It is, above all, a current thought articulated with a broad anti-crisis strategy, but which at no time aims at overcoming the capitalist mode of production. It is an alliance between the fundamental industrial classes and the State, or even a new regime of accumulation and social regulation that extends from 1945 to 1973, characterized by the following aspects: (1) emergence, consolidation, and expansion of monopolism and imperialism; (2) mass production; (3) mass consumption market; (4) specialization and standardization of work; (5) consolidation of positivist rationality; and (6) welfare state (Montaño and Duriguetto 2011, pp. 149–61).

In this context, Ernest Mandel (1982, pp. 333–34), a Belgian Marxist economist and critic of Keynesianism, classifies the attributions of the State as follows: (1) create general conditions of production that cannot be ensured by the private activities of members of the ruling class; (2) repress any threat from the dominated classes or particular fractions of the dominant classes to the current mode of production; and (3) integrate the dominated classes through the ideological reproduction of the dominant class. Such contributions help to understand, beyond appearances, the consequences involved in the adoption of a social practice based on Keynesian thought: the maintenance of capitalist relations corresponds to such a practice.

It is possible to identify in the studies of this group some elements that reinforce this naturalizing interpretation of the capitalist mode of production. One example is the recommendation by Van Barneveld et al. (2020) to rebuild national and global consensus to reimagine the social contract, placing sustainability, equity, and solidarity at the center. Following this ideology, Dean et al. (2021) argue, as a proposal for Australia's post-COVID-19 economic development, for the investment in a 'modern and sustainable industrial policy', with a focus on the use of renewable energy, capable of ensuring full national production of some products (defense, energy, and health) and expanding the manufacturing sector. With regard to productive and reproductive work performed by women, Jenkins and Smith (2021) suggest rethinking investment priorities, directing them to areas such as care for the elderly and disabled, and investment programs to adapt local housing, neighborhoods, and retail centers, and also suggest wage subsidies (via State in an anti-cyclical policy; hence anti-crisis) for work at home, contractual protection against "piecework", and limiting long working hours for women.

The second group is characterized by the adoption of 'post-Keynesianism' as an analytical paradigm. Post-Keynesianism can be understood as a heterodox current of economic thought outlined from the 1970s that is based on principles of the non-neutrality of money, non-ergodicity of the world (that is, inability to predict future events), and uncertainty (Carvalho 2020). The authors Colley et al. (2021) elaborate their work from the 'public service bargain', a theory that understands political–administrative relations as exchanges of benefits and advantages, aligned with the post-Keynesian ideology, especially for its interpretation of social reality focused on the pursuit of ideal institutional structures of a developed capitalism.

Reinforcing this thought, Fazzari and Needler (2021) highlight the ability of theory and heterodox evidence to identify the reasons for the current crisis, due to the concept of stagnation presented by them, and point to the need for an effective macro-economic policy to ensure social equity. It is noted, despite the progressive rhetoric, that the analytical tools and the theoretical framework of post-Keynesianism prove to be insufficient to design ways of overcoming the current capitalist crisis. Not even the most optimistic hypotheses of the projections made by Canelli et al. (2021) for the Italian economy, based on a structural macro-econometric modelling, point to an encouraging scenario, concluding that the policies implemented by the European Union are not sufficient to eliminate the negative effects caused by the pandemic of COVID-19.

The third group contains an article representing the 'post-Keynesian Marxist' economics ideology, elaborated by Baines and Hager (2021). The authors use the capital-as-power approach, proposed by Nitzan and Bichler in 2009, who interpreted capital as quantified power and reject labour value theory, asserting 'capitalization' as the key to understanding capitalist society (Cochrane 2010). In other words, contemporary social dynamics are not derived from the appropriation of labour value generated by workers, but from all social relations controlled, influenced, and transformed by inter-capitalist disputes (Cochrane 2010). Such a perspective of analysis limits the authors' (Baines and Hager 2021) opinion of conceiving the COVID-19 crisis as a missed opportunity for policymakers, who could use their fiscal and monetary power to build a more stable and equitable financial system. Such a way out aligns, to some degree, with the position presented by the first group. Ultimately, in proposing the maintenance of financial capital, their political–ideological positioning reveals itself as favourable to the maintenance of capitalist relations, even if qualitatively different, envisioning as possible a "humanized exploitation".

The fourth group is composed of 13 articles that structure their analysis from historical–dialectical materialism, being classified as 'Marxist'. In general, this group offers an expanded understanding of the current capitalist crisis and the COVID-19 pandemic, placing them as sides of the same coin, or rather, as members of the same totality. Therefore, they advance the understanding of the current historical moment in relation to the studies gathered in the other groups, precisely because they guide their analyses from the dialectical logic that has contradiction as its principle. It is impressive that only Melim and Moraes

(2021) and Carnut et al. (2020) make clear the use of dialectical historical materialism as a method of analysis. Thus, the classification of the other articles in this group was based on the identification of the theoretical framework used to discuss the data presented. One of the common positions among the studies is the conception of the working class as a revolutionary subject necessary for the rupture of the capitalist order.

The fifth group was identified as 'neo-Marxist'. It contains articles that present heterodox conceptions of Marxism, with varying degrees of distancing from classic Marxist assumptions and incorporating new theoretical frameworks. In Kiliç (2020) we see the use of long wave theory, referenced mainly in Schumpeter (1939) and Mandel (1982): the first inserted in the neoclassical economic tradition, and the second in the neo-Marxist current. The authors O'Keeffe and Papadopoulos (2021) used critical discourse analysis, referenced in the post-structuralist discourse theory interpreted by Smith–Carrier. It must be remembered that post-structuralist discourse theory stems from Laclau and Mouffe's formulations heavily influenced by the perspective of post-modern social theory (which advocates that all classical social theories—including Marxist—should be deconstructed for basing their knowledge on a purported universalism, of a totalizing and, above all, rational–phallocentrist character) (Carnut 2019).

The last group, represented by the work of Banfield (2021), took the 'neoclassical' perspective. This current of economic thought is built on the deductive method of David Ricardo and the utilitarianism of Jeremy Bentham, incorporated by Alfred Marshall in *Principles of Economics* (1890) (Milonakis 2020). The basic principles of this theory are methodological individualism, rationality, the abstract/deductive method, the price equilibrium theory, and the subjectivist theory of value (Milonakis 2020). After the Second World War, these principles were restructured in the process of 'formalist revolution', in which the deductive model stands out as a way of validating economic theories. It is no wonder that Banfield (2021) does not question the austere scenario posed to higher education institutions in his text on the role of COVID-19 in remote teaching, but bends to the limits of the logic of cost savings by analyzing the possibilities within the order to sustain access to curricular contents for students.

### 3.4. Methodological Aspects of the Articles

Regarding the methodology used by the authors, it was identified that most of them used literature reviews to analyze their objects of study (Van Barneveld et al. 2020; Carnut et al. 2020; Kiliç 2020; Saad-Filho 2020; Alencar 2021; Baines and Hager 2021; Banfield 2021; Benanav 2021; Bispo and Caldeira 2021; Canelli et al. 2021; Colley et al. 2021; Dean et al. 2021; Jenkins and Smith 2021; Lima 2021; Lust 2021; Ribeiro 2021; Sabino and Alves 2021; Silva and Silva 2021; Storm 2021; Vernengo and Nabar-Bhaduri 2020; Paulsson and Koglin 2022). The literature retrieved in each article found convergences depending on the affiliation of its authors to each analytical paradigm.

Another group used data from the Continuous National Household Sample Survey (PNADC), carried out by the Brazilian Institute of Geography and Statistics (IBGE), to analyze the labour market during the COVID-19 pandemic (Lima et al. 2021; Mattei and Heinen 2021; Pereira and Puchale 2021). Despite starting from different analytical perspectives, as already exposed, the studies pointed out similar trends, such as an increase in the rate of informal work and an increase in the underemployment of workers during the new coronavirus pandemic.

A third group used Keynesian-inspired explanatory models to analyze the economic situation in the context of the COVID-19 pandemic (Araujo et al. 2021; Bortz et al. 2020; Canelli et al. 2021; Fazzari and Needler 2021). A common feature among them is the use of a deductive (mathematical) model as a form of scientific validity, inherited from the formalist phase of neoclassical economics. This way of interpreting the economy, according to Milonakis (2020, p. 207) resumes the idealist vision of a world inhabited by "perfectly rational and selfish human beings, forming rational expectations about the future and exchanging their products in perfectly competitive and efficient markets", and

finds its justification in the very nature of the highly financialized capitalist system and the ideological need to assert itself.

A fourth group is composed of two studies from the field of language (O'Keeffe and Papadopoulos 2021; Żuk and Żuk 2022). In Żuk and Żuk (2022), the material for analysis comes from interviews in focus groups with workers. O'Keeffe and Papadopoulos (2021) examine the political discourses of government authorities for the announcement of recovery plans in the face of the COVID-19 pandemic. The fundamental difference between both is that the latter have a defined and explicit methodology (critical discourse analysis).

*3.5. Elements Contrary to the Law of the Tendency of the Rate of Profit to Fall Identified*

According to Marx (2017b), the fall in the general rate of profit is neither greater nor faster because there are counteracting influences. Analyzing the forms of counter-tendency to LTQTL presented in the reviewed articles, it was possible to identify them with some of the more general causes of the weakening of the fall in the general rate of profit described by Marx (2017b).

Ways to increase the degree of exploitation of labour were presented in 22 articles. Among them, it is possible to perceive the description of forms of exploitation of the working class investigated by Marx (2017a): (a) increased extension of the working day (Van Barneveld et al. 2020; Bispo and Caldeira 2021; Colley et al. 2021; Fazzari and Needler 2021; Sabino and Alves 2021; Sant'ana and Montoya 2021; Silva and Silva 2021; Żuk and Żuk 2022); (b) increased labour productivity (Ribeiro 2021; Sabino and Alves 2021; Sant'ana and Montoya 2021; Paulsson and Koglin 2022); (c) increased labour intensity (Van Barneveld et al. 2020; Bispo and Caldeira 2021; Jenkins and Smith 2021; Ribeiro 2021; Sant'ana and Montoya 2021; Żuk and Żuk 2022); and (d) decrease in variable capital (wage) to levels below the value of the labour power (Van Barneveld et al. 2020; Carnut et al. 2020; Araujo et al. 2021; Benanav 2021; Colley et al. 2021; Fazzari and Needler 2021; Jenkins and Smith 2021; Lima 2021; Lust 2021; Vernengo and Nabar-Bhaduri 2020; Paulsson and Koglin 2022; Żuk and Żuk 2022).

The cheapening of the elements of constant capital is observed, above all, in the case of the expansion of remote work during the COVID-19 pandemic. In this movement, the costs of the infrastructure of the workplace, as well as the instruments for its execution, were left under the responsibility of the workers, providing a relative economy for the capitalists (Stewart et al. 2020; Pereira and Puchale 2021; Melim and Moraes 2021). In this sense, Banfield's (2021) proposal for the cheapening of teaching materials for higher education institutions in the United States through technological incorporation stands out again. Another case in point is the workers of transportation and delivery applications, who assume the costs with their own means of work (Sant'ana and Montoya 2021).

Relative overpopulation, in turn, concerns the mass of unemployed workers corresponding to a certain general stage of the development of the productive forces (Marx 2017b, p. 275). In the context of the COVID-19 pandemic, the unemployment rate rose and affected fractions of the working class differently based on conditions such as race and gender: Silva and Silva (2021) recalls that 70.4% of domestic workers around the world suffered from reduced hours worked or lost wages; among this contingent, 92% were women and 63% were black. In Brazil, unemployment was higher among women and black people, following the international trend (Fazzari and Needler 2021; Silva and Silva 2021; Vernengo and Nabar-Bhaduri 2020). It is worth mentioning a caveat made by Mattei and Heinen (2021) about the Brazilian unemployment data, which conceal partial job losses resulting from the processes of the removal of workers from their occupations and reduction in working hours.

As for the increase in shareholder capital, Marx (2017b, p. 279) points out that its composition derives from dividends; that is, from the large or small interest on capital invested in large productive enterprises. As Höfig (2017) summarizes, in shareholder capital "the capital-value is fully duplicated: it exists effectively in the capital immobilized in the process of production and circulation of commodities, and ideally in shareholder

capital". At present, shareholder capital underwent a process of autonomization, largely due to the reconfiguration of the regulatory framework of the financial market since the 1970s (Höfig 2017). During the COVID-19 pandemic, a group of articles described the paths of this increase through purchases of private debt securities and the national treasury by the State; increase in inflation, increase in State public debt; interventions on the exchange rate and financial markets; financing for large companies, some transferred to shareholders as dividends; and transfers and purchases of debt securities by the FED and transfers of funds to commercial banks, used for stock purchases (Šumonja 2020; Araujo et al. 2021; Baines and Hager 2021; Bortz et al. 2020; Canelli et al. 2021; Ribeiro 2021; Storm 2021).

As exposed, the existence of the counteracting elements in the face of the COVID-19 pandemic is conditioned to the role of the State as an instrument of domination of the bourgeoisie, guaranteeing refluxes of profitability. In this sense, Lust (2021, p. 660) highlights that "the State is not only fundamental for the economic reproduction of the system, but also for its social and ecological reproduction".

It is important to emphasize why studying countertrends is fundamental from a practical point of view. The practical aspects related to this recognition of the countertrends study can be described in two points: (1) they serve to identify with greater clarity the spaces of capitalist social relations advancement. These spaces where capitalists act with countertrends are important indicators to identify the main sectors in which capitalists try to recover their profitability, and (2) they serve to demonstrate to the working class the place where the confrontation of capitalist social relations advancement will be conducted in the coming years with greater intensity.

### 3.6. Limitations of This Review

This article, according to the methodology used, sought to synthesize and critically reflect on the knowledge available from the choice of journals and annals of scientific events that publish Marxist scientific content. This choice implied dealing with some obstacles for the retrieval of articles: (a) malfunctioning of the online search engines of some journals; (b) disorganized layout of some magazines; and (c) the large number of sites accessed to systematize the content. This reflects, in part, the academic discredit experienced by counter-hegemonic theoretical dissemination vehicles and the lack of funding faced by them as a way to prevent their circulation.

Additionally, only the articles available for reading in full and free of charge were selected. It was understood that this choice would be the most adequate to guarantee ethical–political coherence with the Marxist critical analysis carried out in this study, valuing the socialization of the elaborated knowledge.

Finally, the content of the study aimed to focus on a specific point of understanding of the capitalist dynamics, and does not disregard the complexity of phenomena that contemporary capitalism presents as the guarantee of certain advantages to specific classes and fractions of classes.

## 4. Final Considerations

Finally, it is possible to say that in view of the loss of profitability, the reviewed articles converge to demonstrate that countertendencies are quickly triggered by capitalists and the State, being justified by the latter due to the health crisis, and taking the latter as a "fatality". Such countertendencies are related to: (1) increased degree of labour exploitation; (2) cheapening of the elements of constant capital; (3) increased relative overpopulation; and (4) increased shareholder capital. Meanwhile, the most vulnerable parts of the working class (women, blacks, and immigrants) were the most precarious, suffering the most from the advance of precarization and 'secondary expropriation'. The advance of destruction and exploitation of natural resources also stands out, putting the possibilities of human existence in the long term in jeopardy.

Still, it is important to restate that, according to this same evidence, these counter-tendencies were already in progress even before the new coronavirus pandemic and were

intensified by it. For this reason, in historical terms, countertendencies cannot be interpreted as measures resulting only from the fight against the COVID-19 pandemic, but rather as ways to protect the capital. Being precise, this means that most of these countertrends were already being adopted, not being directly related to the pandemic period (with exceptions, such as those measures related to remote work, for example). Finally, it is necessary to reaffirm that this evidence grouped together is the result only of the journals that were reviewed, and therefore, they translate, "roughly speaking", a synthesis of the Marxist analysis on the theme—given other journals that were not part of the review, and also the mix of paradigms that are added to the Marxist explanation found in the journals that were reviewed.

**Author Contributions:** The authors (L.C., L.U., Á.M.) declare that they contributed equally to the preparation of the manuscript as to the conception and planning for the analysis and interpretation of the data; the draft's elaboration and content's critical review; and the approval of the final version of the manuscript. All authors have read and agreed to the published version of the manuscript.

**Funding:** This research received no external funding.

**Informed Consent Statement:** Not applicable.

**Data Availability Statement:** Not applicable.

**Conflicts of Interest:** The authors declare that they have no conflict of interest regarding the conception of this work.

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
