# Peer review of "Has the COVID-19 Pandemic Cooled Down or Stimulated the Countertendencies of Capital? A Critical Review"

_economies, doi:10.3390/economies11050148_

Round 1

Reviewer 1 Report

An informative literature review. It covers many key texts, although I notice some important omissions:
Sandipan Baksi, “Epidemics and Capitalism: Accurate Identification of Contradictions”;
Tithi Bhattacharya and Gareth Dale, “General Tendencies, Possible leaps”;
Vincent Lyon-Callo, “COVID and Capitalism: A Conversation with Richard Wolff”;
Dimitris Fasfalis, “Marx in the era of pandemic capitalism”;
Anitra Nelson, “COVID-19: Capitalist and postcapitalist perspectives”

The title isn’t ideal. Better would be to make clear that it is not the pandemic that is cooling/intensifying the counter-tendencies, but that it has prompted or catalysed these tendencies. However, of course titles should be pithy, and readers will understand the compression.

Reviewer 2 Report

This is a well written paper, with some robust argumentation and interesting findings. Paper’s major merit seems to be the extended and well-informed updated bibliography. Although interesting may be, I fear that the article needs significantly more work before it could be published.

First, the author should incorporate in the introduction the following two articles that give a panoptical view of the current dynamics of capitalism: 1. Giorgos Meramveliotakis (2022) Understanding Money & Credit in Contemporary Capitalism: Back to the Future of Marx's Theory of Fetishism and Alienation, Critique, 50:2-3, 307-324, DOI: 10.1080/03017605.2022.2123605, 2. Munger, M. C., & Villarreal-Diaz, M. (2019). The road to crony capitalism. The Independent Review, 23(3), 331-344.

Second, the author’s assertion in the final remarks that “in historical terms, countertendencies cannot be interpreted as measures resulting from the fight against the COVID-19 pandemic, but rather as ways to save capital” consists of a rather crude argumentation, since one could equally argue that in the specific historical context the measures taken was indeed to address COVD-19 pandemic’s severe consequences.

Reviewer 3 Report

The paper under consideration is well written and logically structured. Its reasoning is clear. It proposes an interesting, rarely presented in economic journals, perspective (a Marxist approach) to analyse measures taken during the Covid-19 pandemic.

However, in my opinion, the authors did not maintain sufficient scientific objectivity, which reduces the scientific value of the text. It seems that the content of the study is subordinated in advance to the initial assumption: capitalism is only a bad system with no advantages. Removing such minor ideological accretions from the text would increase its credibility. However, I want to clearly emphasize that these are my subjective impressions after reading the text.

In addition, before publication, I would recommend the following:

1.       When formulating the main objective of the analysis, it would be reasonable to link it to the question posed in the title of the study.

2.       Expanding the conclusion section by referring to the main limitations of the study.

3.       From the reader's point of view, it would also be interesting to indicate the practical implications of the analyses.

4.       Section: Central aspects of studies….: I would change the order of the parts (in terms of deductive reasoning from general ideas to specific conclusions) to the following: 1. Cases studied 2. Case analysis context 3. Analysis matrices 4. Methodological aspects 5. Elements contrary…..

5.       Some references seem to be missing in the content of the paper: e.g.  According to Manzano…., As Mendes and Carnut point out…., As Pereira and Pererira-Pereira emphasize ….

Round 2

Reviewer 2 Report

Authors accommodate all of my comments. I strongly propose publication.